# DPAF: Image Synthesis via Differentially Private Aggregation in Forward Phase

## Abstract

Differentially private synthetic data is a promising alternative for sensitive data release. Many differentially private generative models have been proposed in the literature. Unfortunately, they all suffer from the low utility of the synthetic data, especially for high resolution images. Here, we propose DPAF, an effective differentially private generative model for high-dimensional image synthesis. Unlike previous methods, which add Gaussian noise in the *backward* phase during model training, DPAF adds differentially private feature aggregation in the *forward* phase, which brings advantages such as reducing information loss in gradient clipping and low sensitivity to aggregation. Since an inappropriate batch size has a negative impact on the utility of synthetic data, DPAF also addresses the problem of setting an appropriate batch size by proposing a novel training strategy that asymmetrically trains different parts of the discriminator. We extensively evaluate different methods on multiple image datasets (up to images of $128 \times 128$ resolution) to demonstrate the performance of DPAF.

## 1 Introduction

Training deep neural networks (DNNs) requires a large amount of high-quality data. Unfortunately, much valuable data is privacy sensitive, and direct release of the data becomes infeasible. Synthetic data has been proposed as a means to overcome the above difficulty. In particular, synthetic data from generative models can have the same statistical information as the original data, and therefore leads to great data utility. Synthetic data is usually assumed to be decoupled from the original data and therefore implies privacy. However, recent studies reveal that the generative models still leak privacy information due to, for example, membership inference attacks (MIAs) (Chen et al., 2020b; Hayes et al., 2019; Hilprecht et al., 2019).

Differentially private deep learning (DPDL), a variant of deep learning (DL) with the differential privacy (DP) (Dwork & Roth, 2014), can be used to provably preserve the privacy of DL models. For example, Abadi et al. (2016) propose differentially private stochastic gradient descent (DPSGD) to train a DP classifier by adding the Gaussian noise to the clipped gradient. Subsequent works apply DPSGD to generative models such as generative adversarial networks (GANs) and diffusion models (Dhariwal & Nichol, 2021; Ho et al., 2020) to derive synthetic data with DP. However, DPSGD has negative (Bagdasaryan et al., 2019) effects on the model utility.

In this work, we aim to design a DPGAN for image synthesis by relying on DPSGD. Our goal is to generate high-dimensional images in a DP manner so that the downstream classification task can have high classification accuracy. It has been widely known that the utility of DPSGD-based DP-GANs is degraded due to two factors, the information loss in gradient clipping and DP noise. In this sense, our proposed techniques are designed to reduce both the information loss (by minimizing the size of the gradient vector) in gradient clipping and DP noise (by minimizing the global sensitivity).

In particular, we perform a DP feature aggregation in the forward phase during the training of a DNN. The postprocessing of DP ensures that the number of dimensions of the gradient vector can be effectively reduced. This greatly reduces information loss from gradient clipping in DPSGD. On the other hand, choosing a large batch size is a straightforward way to reduce the negative impact of DP noise, because the DP noise can be easier to cancel each other out. Unfortunately, a large batch size leads to much fewer updates of the discriminator, and thus the training cannot converge given a fixed number of epochs. In addition, due to the feature aggregation in our proposed method, the

large batch size also easily makes features indistinguishable, which is detrimental to the utility of the synthetic data and, in turn, favors a small batch size instead. We address this dilemma through an elaborate design of the training strategy.

**Contribution.** The contributions are summarized below.

- We propose DPAF (**D**ifferentially **P**rivate **A**ggregation in **F**orward phase), an effective generative model for differentially private image synthesis. DPAF supports the conditional generation of high-dimensional images.

- We propose a novel framework to enforce DP during GAN training. In particular, we propose to place a differentially private feature aggregation (DPAGG) in the forward phase. Together with a simplified instance normalization (SIN), DPAGG can not only have a natural and low global sensitivity, but also significantly reduce the dimensionality of the gradient vector. We also have a novel design of asymmetric model training, which solves the dilemma that a small batch cannot effectively reduce the DP noise, but a large batch will make features indistinguishable.

- We formally prove the privacy guarantee of DPAF. Furthermore, our extensive experiment results confirm DPAF's generative capability for large-scale images.

## 2 BACKGROUND KNOWLEDGE AND RELATED WORK

**Differential Privacy (DP).** $(\varepsilon, \delta)$-differential privacy (Dwork & Roth, 2014), $(\varepsilon, \delta)$-DP, is the *de facto* standard for data privacy, where the *privacy budget* $\varepsilon > 0$ measures the *privacy loss* and $\delta \in [0, 1]$ is the probability of violating DP.

**Definition 1.** *An algorithm $\mathcal{M}$ is $(\varepsilon, \delta)$-DP if $Pr[\mathcal{M}(\mathcal{D}) \in \mathcal{S}] \leq e^{\varepsilon} Pr[\mathcal{M}(\mathcal{D}') \in \mathcal{S}] + \delta$, for all $\mathcal{S} \subseteq Range(\mathcal{M})$ and for any neighboring datasets $\mathcal{D}$ and $\mathcal{D}'$.*

DP can be achieved by applying the Gaussian mechanism $G_{\sigma}$, where the zero-mean Gaussian noise with appropriate variance is added to the algorithm output. Specifically, given a function $f$, $G_{\sigma} \circ f(x) \triangleq f(x) + N(0, \sigma^2)$ satisfies $(\varepsilon, \delta)$-DP for all $\varepsilon < 1$ and $\sigma > \sqrt{2 \ln(1.25/\delta)} \Delta_{2,f}/\varepsilon$, where the $\ell_2$-sensitivity of $f$ is defined as $\Delta_{2,f} \triangleq \max_{\mathcal{D}, \mathcal{D}'} ||f(\mathcal{D}) - f(\mathcal{D}')||_2$ for neighboring $\mathcal{D}$ and $\mathcal{D}'$.

DP has the following useful characteristics. First, repeated access to sensitive data leads to an accumulation of privacy loss (e.g., sequential composition (Dwork & Roth, 2014; Doroshenko et al., 2022; Abadi et al., 2016)). Second, DP is not affected by postprocessing. Formally, $g \circ \mathcal{M}$ for any data-independent mapping $g$ still satisfies $(\varepsilon, \delta)$-DP, given that $\mathcal{M}$ is $(\varepsilon, \delta)$-DP.

**DP Generative Models.** DPSGD (Abadi et al., 2016) is a popular technique to train a DP model. Essentially, DPSGD clips the gradient and adds the Gaussian noise to it. More details about DPSGD can be found in Appendix A.1. Gradient clipping in DPSGD is beneficial in reducing global sensitivity, but leads to a dramatic loss of information. Therefore, several works propose techniques such as adaptive clustering to reduce the information loss (Thakkar et al., 2021; McMahan & Andrew, 2018; Zhang et al., 2018; Yu et al., 2021; Nasr et al., 2020). One can also find many DPSGD-based DPGANs such as GANobfuscator (Xu et al., 2019), GS-WGAN (Chen et al., 2020a), Private-GANs (Bie et al., 2023), and (Xie et al., 2018). DPSGD can also be used on autoencoders (Jiang et al., 2022; Pfitzner & Arnrich, 2022; Takagi et al., 2021) and diffusion models (Lin et al., 2023; Ghalebikesabi et al., 2023; Dockhorn et al., 2023). Another technique often used for DP generative models is PATE (Papernot et al., 2017; 2018). Through PATE, PATE-GAN (Jordon et al., 2019) uses the noisy labels on the samples generated by the generator to train both the generator and the discriminator based on the PATE framework. Similar designs include G-PATE (Long et al., 2021) and DataLens (Wang et al., 2021). On the other hand, DP-MERF (Harder et al., 2021) trains the generator by considering the maximum mean discrepancy (MMD) over random feature representations of kernel mean embeddings for both the data and generator distributions. Some of subsequent works include DP-MEPF (Harder et al., 2023), DP-NTK (Yang et al., 2023), DP-HP (Vinaroz et al., 2022), DP-Sinkhorn (Cao et al., 2021), and PEARL (Liew et al., 2022). Finally, DPGEN (Chen et al., 2022) and DPDC (Zheng & Li, 2023) rely on Langevin Markov chain Monte Carlo (MCMC) and dataset condensation Zhao et al. (2021), respectively.

**Goal.** Our goal is to develop a DPGAN with a data high utility. More specifically, we assume that only the generator of a GAN is released for data synthesis. In other words, an attacker's access to the generator should not lead to privacy leakage of training samples.

## 3 DPAF

Here, we present DPAF (**D**ifferentially **P**rivate **A**ggregation in **F**orward phase), an effective generative model for differentially private image synthesis. The workflow of DPAF is illustrated in both Algorithm 1 in Appendix A.2 and Figure 1. A notation table can be found in Table 9 in Appendix A.3.

The high-level idea of DPAF in reducing information loss during gradient clipping, preserving gradient structure, improving robustness against DP noise, and lowering global sensitivity is illustrated in Appendix A.4. In particular, DPAF is trained by using transfer learn-

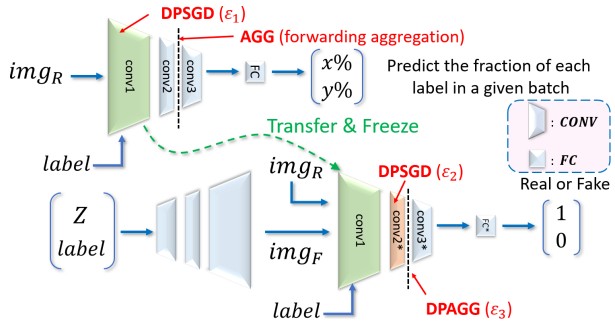

Figure 1: The model architecture of DPAF.

ing. Hence, similar to transfer learning, DPAF has two phases, training a classifier first and training a GAN, both in the DP manner.

Though DPAF uses transfer learning as a subroutine to improve utility, it does not use warm start (Zhang et al., 2018), a technique to improve utility by exploiting extra data with similar data distribution, because it is not common to find such data for arbitrary sensitive datasets.

### 3.1 TRAINING A CLASSIFIER BEFORE TRANSFER LEARNING

The architecture of the classifier $C$ in DPAF before transfer learning is shown in the upper part of Figure 1. The classifier $C$ is identical to the discriminator of cDCGAN (in fact, a standard convolutional neural network (CNN), see Appendix A.5), where there are three parts of convolutional layers (conv1, conv2, and conv3) and some fully connected (FC) layers, except that an ordinary aggregation layer is added between conv2 and conv3. Each of conv1, conv2, and conv3 may consist of multiple convolutional layers, depending on the input image size. We follow the common setting that the height/width of the feature maps in the next convolutional layer is half that of the current convolutional layer.

The ordinary CNN is designed to classify inputs. However, because we introduce an aggregation layer (AGG) between conv2 and conv3, which aggregates multiple normalized features in a batch, obviously the CNN can no longer output the predicted class for a given input image and label. Instead, $C$ here is designed to predict the percentage of each class in a given batch, as shown in Figure 1. To achieve the above goal, $C$ is trained with the labeled sensitive data (img$_R$ in Figure 1) through the mean square error (MSE) loss function, $\mathcal{L}_{\text{MSE}}$. Afterward, in one round of the backpropagation, SGD is applied to conv2, conv3, and FC for the update of the corresponding parameters, while the DPSGD with the privacy budget $\varepsilon_1$, DPSGD($\varepsilon_1$), is applied to conv1, because only conv1 will be recycled to be used after transfer learning. Corollary 1 in Appendix A.6 proves that conv1 satisfies DP. Lines 2~4 in Algorithm 1 correspond to the above training procedures. After training, conv2, conv3, and FC are discarded and will not be released.

### 3.2 TRAINING A DPGAN AFTER TRANSFER LEARNING

In this phase, we aim to train a DPGAN such that a generator satisfying DP can be released. GAN is known to be composed of two parts, a generator $G$ and a discriminator $D$. In DPAF, the architecture of $D$ is identical to $C$, as shown in the bottom part of Figure 1. The conv1 in $C$ is transferred to be the conv1 in $D$; i.e., $C$ and $D$ share the same conv1 (Line 5 in Algorithm 1). Such a conv1 does not

leak privacy as it is trained by DPSGD, though conv2, conv3, and FC are trained by SGD[1] (see the formal proof in Theorem 1 and Corollary 1 in Appendix A.6). Here, conv2*, conv3*, and FC* are randomly initialized. Unlike $C$, where an AGG is placed between conv2 and conv3, a DP feature aggregation layer with the privacy budget $\varepsilon_3$, DPAGG($\varepsilon_3$), is placed between conv2* and conv3* of $D$. The architecture of $G$ is a reverse of $D$ without DPAGG($\varepsilon_3$).

Training the DPGAN in DPAF is similar to training an ordinary GAN; i.e., we iteratively train $D$ first and then the $G$ until the convergence. $D$ takes as input the sensitive images (img$_R$ in Figure 1), synthetic images (img$_F$ in Figure 1), and the label. Given that conv1 is frozen during the training due to transfer learning, $D$ is trained to differentiate between real and synthetic images. The general guideline of training $D$ is that after binary cross entropy (BCE) loss function $\mathcal{L}_{\text{BCE}}$ is calculated on $D$ with DPAGG($\varepsilon_3$), conv3* and FC* can be updated via SGD because such an update of conv3* and FC* still satisfies the DP according to the postprocessing of DPAGG($\varepsilon_3$). Consider $\tilde{D}$ as the discriminator $D$ that replaces DPAGG($\varepsilon_3$) with AGG. In addition, another BCE loss function $\mathcal{L}'_{\text{BCE}}$ is calculated on $\tilde{D}$. conv2* will be updated by using DPSGD($\varepsilon_2$) based on $\mathcal{L}'_{\text{BCE}}$. Lines 7~13 in Algorithm 1 correspond to the training of $D$. The details about how $\mathcal{L}_{\text{BCE}}$ and $\mathcal{L}'_{\text{BCE}}$ are related to our proposed asymmetric training will be described below.

Consider $\hat{D}$ as the discriminator $D$ without DPAGG($\varepsilon_3$); i.e., $\hat{D}$ can be seen as a standard CNN. After updating $D$ $n_{\text{critic}}$ times, the BCE loss function $\mathcal{L}''_{\text{BCE}}$ is calculated on $\hat{D}$ and is backpropagated to update $G$ through SGD, where $n_{\text{critic}}$ is the number of critic iterations per generator iteration for better training (Arjovsky et al., 2017). The design of skipping the aggregation in $D$ when training $G$ can be attributed to the fact that we aim to learn how to generate a single image, instead of a mix of images. Note that $\mathcal{L}_{\text{BCE}}$, $\mathcal{L}'_{\text{BCE}}$, and $\mathcal{L}''_{\text{BCE}}$ all work on the same $D$, but depending on which part of $D$ needs to be updated, different components of $D$ are ignored. Lines 14~16 in Algorithm 1 correspond to the training of $G$.

**The Design of DPAGG.** AGG can be implemented via two steps. First, the normalized feature maps are concatenated as a feature vector. Second, the feature vectors from different samples in a batch are aggregated. We can have DPAGG when we apply the Gaussian mechanism to the aggregated feature vector derived from AGG. Below we describe how our instance normalization works.

Inspired by (Ulyanov et al., 2017; 2016), we propose to use a simplified instance normalization (SIN) to not only ensure the balance of feature values but also, more importantly, derive a bound of the global sensitivity of AGG. SIN can be formulated as follows.

$$\mu_{i_1 i_2} = \frac{1}{HW} \sum_{i_3=1}^{H} \sum_{i_4=1}^{W} x_{i_1 i_2 i_3 i_4}, \sigma^2_{i_1 i_2} = \frac{1}{HW} \sum_{i_3=1}^{H} \sum_{i_4=1}^{W} \left( x_{i_1 i_2 i_3 i_4} - \mu_{i_1 i_2} \right)^2, \widehat{x_{i_1 i_2 i_3 i_4}} = \frac{x_{i_1 i_2 i_3 i_4} - \mu_{i_1 i_2}}{\sqrt{\sigma^2_{i_1 i_2}}}, \quad (1)$$

where $\mu_{i_1 i_2}$ is the mean of feature map $X_{i_1 i_2}$, $\sigma^2_{i_1 i_2}$ is the variance of $X_{i_1 i_2}$, $i_1$ is the index of the image in the batch, $i_2$ is the feature channel (color channel if the input is an RGB image), $H$ is the height of the feature map, $W$ is the width of the feature map, $x_{i_1 i_2 i_3 i_4} \in \mathbb{R}$ is an element of feature map $X_{i_1 i_2}$, $\widehat{x_{i_1 i_2 i_3 i_4}}$ is the new value of $x_{i_1 i_2 i_3 i_4}$ after SIN. One can easily see that SIN is different from ordinary instance normalization (IN) in that SIN does not have learnable parameters, center and scale (Ulyanov et al., 2017; 2016). Concretely, each normalized feature map (through SIN) is guaranteed to have the same $\ell_2$-norm $p$, where $p \times p$ is the size of feature map.

Before applying the Gaussian mechanism to the AGG output, we calculate the $\ell_2$-sensitivity $\Delta_{2,\text{AGG}}$ of the AGG below.

$$\Delta_{2,\text{AGG}} = \sqrt{\sum_{x \in X_{11}} (\frac{x - \mu_{11}}{\sigma_{11}})^2 + \cdots + \sum_{x \in X_{1m}} (\frac{x - \mu_{1m}}{\sigma_{1m}})^2} = \sqrt{\frac{1}{\sigma^2_{11}} \sum_{x \in X_{11}} (x - \mu_{11})^2 + \cdots + \frac{1}{\sigma^2_{1m}} \sum_{x \in X_{1m}} (x - \mu_{1m})^2}$$

$$= \sqrt{\sum_{j=1}^{m} \left[ \frac{\|X_{1j}\|}{\sum_{x \in X_{1j}} (x - \mu_{1j})^2} \sum_{x \in X_{1j}} (x - \mu_{1j})^2 \right]} = \sqrt{\|X_{11}\| + \cdots + \|X_{1m}\|} = \sqrt{p^2 + \cdots + p^2} = \sqrt{m} p, \quad (2)$$

---

[1]GS-WGAN (Lines 17 and 19 in Algorithm 1 of (Chen et al., 2020a)), G-PATE (Lines 10 and 14 in Algorithm 1 of (Long et al., 2021)), and DataLens (Lines 10 and 14 in Algorithm 1 of (Wang et al., 2021)) have a similar design, where some parts of the model are trained by SGD but eventually discarded while the remaining parts are trained by DPSGD for the eventual release.

| batch # | batch 1 | batch 2 | batch 3 | batch 4 | batch 5 | batch 6 | batch 7 | batch 8 | batch 9 | batch 10 |
|---------|---------|---------|---------|---------|---------|---------|---------|---------|---------|----------|
| update G? | | | G | | | G | | | G | |
| update D? | conv3* FC* | conv3* FC* | conv3* FC* | conv3* FC* | conv3* FC* | conv3* FC* | conv3* FC* | conv3* FC* | conv3* FC* | conv3* FC* |
| | | | | | | | | conv2* | | |

Figure 2: The asymmetric training with $\mu = 8$ and $n_{\text{critic}} = 3$.

where $||X||$ is the size of feature map $X$, $m$ is the number of feature maps, $x$ is an element of feature map $X_{1j}$ for $j = 1, 2, \cdots, m$. In the above calculation of $\Delta_{2,\text{AGG}}$, we consider batch size 1 because we aim to know the amount of difference to which a single sample in the batch contributes.

Depending on the tasks, the normalization can be placed in a different position or even multiple layers (Jin et al., 2020; Huang & Belongie, 2017) for better training. We find that in addition to offering a fine-grained control of features similar to computer vision tasks, SIN in DPAF plays a unique role in bounding $\ell_2$-sensitivity, though SIN is an easy modification of IN. Note that we apply SIN to only those feature maps just before DPAGG. Such a design is supported by our experiments that applying SIN in all the layers before DPAGG, in turn, degrades the utility because, unlike IN, SIN lacks the learnable parameters.

**Asymmetric Training of $D$ in DPAF.** In fact, AGG asks for a smaller batch size because, otherwise, the features will be mixed and cannot be recognized. Nevertheless, a smaller batch size, in turn, is harmful to DPSGD because the DP noise will make a greater impact on the gradient. Hence, DPAF prefers a larger batch size from the DPSGD point of view. We propose an asymmetric training strategy to resolve the contradicting requirements of setting a proper batch size. In essence, in the asymmetric training, when training $D$, we update conv3* and FC* through SGD for every iteration but update conv2* through DPSGD($\varepsilon_2$) for every $\mu$ iterations, as shown in Figure 2. $\mu$ is called *asymmetry multiplier* because it determines the ratio of the privacy budget for conv2* and the budget for both conv3* and FC*.

More specifically, for each batch, $\mathcal{L}_{\text{BCE}}$ is calculated by feeding the samples to $D$ and then we update conv3* and FC* through SGD. At the same time, for the $i$-th batch with $\mu \mid i$, $\mathcal{L}'_{\text{BCE}}$ is calculated by feeding all of the samples from the $\mu$ latest batches to $\tilde{D}$ and then we update conv2* through DPSGD($\varepsilon_2$). An example of asymmetric training is shown in Figure 2. Here, we make an important observation that updating conv2* through DPSGD($\varepsilon_2$) for every $\mu$ iterations virtually increases batch size $\mu$ times for conv2*.

Given that $\mu$ controls the batch size for updating conv2*, a natural question that arises is whether $\mu$ can be increased arbitrarily. In fact, we cannot arbitrarily increase $\mu$ because the increased $\mu$ also leads to a less frequent update of conv2*, which may, in turn, degrade the utility.

Several questions remain for DPAF; for example, why not eliminate conv2* and why not have more layers for conv2* (or conv3*)? Appendix A.5 answers the above questions. In addition, Appendix A.5 further describes the best choice of cGAN and optimizes the position of DPAGG. Finally, Theorem 6 in Appendix A.6 formally proves that DPAF satisfies DP.

## 4 EXPERIMENT EVALUATION

### 4.1 EXPERIMENT SETUP

**Dataset.** In our experiments, we considered MNIST, Fashion MNIST (FMNIST), CelebA, and FFHQ datasets. Both MNIST and FMNIST have $28 \times 28$ grayscale images. CelebA contains colorful celebrity images of different sizes. In our experiments, we rescaled all of CelebA images into $64 \times 64$ colorful images. Based on CelebA, we created two more datasets, CelebA-Gender and CelebA-Hair, where the former is for binary classification with gender as the label and the latter is for multiclass classification dataset with hair color (black/blonde/brown) as the label. FFHQ contains

70,000 colorful facial images of $128 \times 128$ with gender as label[2] and we created FFHQ-Gender dataset for binary classification.

**Baselines.** We considered the baseline methods, GS-WGAN (Chen et al., 2020a), DP-MERF (Harder et al., 2021), P3GM (Takagi et al., 2021), DataLens (Wang et al., 2021), G-PATE (Long et al., 2021), DP-Sinkhorn(Cao et al., 2021), DP-HP (Vinaroz et al., 2022), Nonlinear DPDC (NDPDC) (Zheng & Li, 2023), and PEARL (Liew et al., 2022). The implementation of all the baselines is based on the official code (see Appendix A.7). Though the official code is not available online, we communicated with the authors of PEARL to have a copy.

**Evaluation Metrics.** Given two levels of privacy guarantee, $(1, 10^{-5})$-DP and $(10, 10^{-5})$-DP, we aim to evaluate the utility of DP image synthesis. The utility can have two dimensions; i.e., the classification accuracy and the visual quality. In the former case, we calculate the predicting accuracy of the classifier trained by synthetic images and tested by real images. The architecture of the classifier used in our experiment is the same as the one used in GS-WGAN, G-PATE, and DataLens and is shown in Figure 7 in Appendix A.8. We conducted necessary modifications on the code for DP-MERF, P3GM, DP-Sinkhorn, DP-HP, NDPDC, and PEARL to derive their accuracies under the same setting. This explains the inconsistency between the results in this paper and the results reported in the original papers. On the other hand, in the latter case, we display the synthetic images for visualization and report Fréchet inception distance (FID).

| | $\varepsilon$ | GS-WGAN | DP-MERF | P3GM | G-PATE | DP-Sinkhorn | DataLens | DP-HP | NDPDC | PEARL | DPAF |
|---|---|---|---|---|---|---|---|---|---|---|---|
| MNIST | 1 | 0.143 | 0.637 | 0.737 | 0.588 | 0.654 | 0.712 | 0.742 | 0.537 | **0.782** | 0.501 |
| | 10 | 0.808 | 0.674 | 0.798 | 0.809 | **0.832** | 0.807 | 0.744 | 0.360 | 0.788 | 0.748 |
| Fashion- | 1 | 0.166 | 0.586 | **0.722** | 0.581 | 0.564 | 0.648 | 0.651 | 0.594 | 0.683 | 0.543 |
| MNIST | 10 | 0.658 | 0.616 | **0.748** | 0.693 | 0.711 | 0.706 | 0.652 | 0.551 | 0.649 | 0.640 |
| CelebA- | 1 | 0.590 | 0.594 | 0.567 | 0.670 | 0.543 | 0.700 | 0.656 | 0.540 | 0.634 | **0.802** |
| Gender | 10 | 0.614 | 0.608 | 0.588 | 0.690 | 0.621 | 0.729 | 0.617 | 0.600 | 0.646 | **0.826** |
| CelebA- | 1 | 0.420 | 0.441 | 0.453 | 0.499 | × | 0.606 | 0.561 | 0.498 | 0.606 | **0.675** |
| Hair | 10 | 0.523 | 0.449 | 0.486 | 0.622 | × | 0.622 | 0.474 | 0.462 | 0.626 | **0.671** |

Table 1: Classification accuracy results under $(1, 10^{-5})$-DP and $(10, 10^{-5})$-DP. We have two ×'s because we failed to modify the code of DP-Sinkhorn and synthesize CelebA-Hair images. The rightmost column shows canonical accuracy.

**Canonical Implementation of DPAF.** Basically, DPAF adds the DP feature aggregation on the basis of cDCGAN. In our canonical implementation, the batch size is 24 for MNIST and FMNIST and 64 for CelebA and FFHQ. The latent vector sampled from the standard Gaussian distribution is of dimension 100. The asymmetry multiplier $\mu = 8$. We also apply gradient compression (Lin et al., 2018) to the per-sample gradient to keep the top 90% values only.

For the privacy budget allocation, the notation $(x_1\%, x_2\%, x_3\%)$ refers to the setting, where conv1, conv2*, and DPAGG have $\varepsilon_1 = \frac{x_1 \cdot \varepsilon}{100}$, $\varepsilon_2 = \frac{x_2 \cdot \varepsilon}{100}$, and $\varepsilon_3 = \frac{x_3 \cdot \varepsilon}{100}$, respectively, given the total privacy budget $\varepsilon$. A similar notation is $(x_1, \times, x_3)$, where both conv1 and DPAGG have a privacy budget $\varepsilon_1 = x_1$ and $\varepsilon_3 = x_3$, respectively, and conv2* has the rest. For example, $(0.1, \times, 0.1)$ means that conv1, conv2, and DPAGG have $0.1$, $9.8$, and $0.1$, respectively, if the total privacy budget is $10$. Thus, *canonical accuracy* means the accuracy from the canonical implementation.

## 4.2 EXPERIMENT RESULTS

### 4.2.1 CLASSIFICATION ACCURACY

Table 1 shows the classification results of DPAF and the other baseline methods. One can see from Table 1 that DPAF outperforms all other baselines for CelebA-Gender and CelebA-Hair, but surprisingly is worse than some of the other baselines for MNIST and FMNIST. This can be attributed to the architecture behind the design of DPAF. In particular, as mentioned in Sections 3.1 and 3.2, our convolutional layers follow the conventional design, i.e., feature maps are shrunk from the current

---

[2]FFHQ labels from `https://github.com/DCGM/ffhq-features-dataset/tree/master/json`.

layer to the next layer, and consequently the number of convolutional layers depends on the size of the input image. For MNIST and FMNIST, as the image size is smaller, there will be fewer convolutional layers in our canonical design, thus limiting the generative capability of the generator, given the conventional design that $G$ is the inverse of $D$ (e.g., PGGAN (Karras et al., 2018)). On the contrary, the larger image size implies more convolutional layers in DPAF, which strengthens the generative capability. In this sense, our design also suggests the potential of DPAF to synthesize images with higher resolutions, since a generator with more layers can be employed. Such an argument is partially supported by the experimental results in Section 4.2.3.

| | $\varepsilon$ | DataLens | DP-HP | PEARL | DPAF |
|---|---|---|---|---|---|
| CelebA- | 1 | 298 | 352 | 303 | **285** |
| Gender | 10 | 320 | 341 | 302 | **298** |
| CelebA- | 1 | × | 339 | 338 | **301** |
| Hair | 10 | × | 340 | 337 | **298** |

Table 2: The comparison of FIDs for $64 \times 64$ CelebA.

| | $\varepsilon$ | PEARL | DPAF |
|---|---|---|---|
| FFHQ- | 1 | $0.441 \pm 0.019$ | $\mathbf{0.567} \pm 0.038$ |
| Gender | 10 | $0.511 \pm 0.027$ | $\mathbf{0.646} \pm 0.004$ |

Table 3: Accuracy comparison for $128 \times 128$ FFHQ-Gender.

### 4.2.2 VISUAL QUALITY

We present the results of the visual quality evaluation in Figure 3. In particular, the DPAF-synthesized images appear more realistic and capture more facial features, such as eyes, lips, and facial shape, compared to the uniform faces generated by DP-Sinkhorn and the highly noisy faces generated by the other baselines. Such a gain is due to the use of SIN and our design of DPAGG, i.e., the aggregated feature is more robust to the DP noise and better able to discriminate features after training. We also present the quantitative results in Table 2. We can see that DPAF achieves the lowest FIDs.

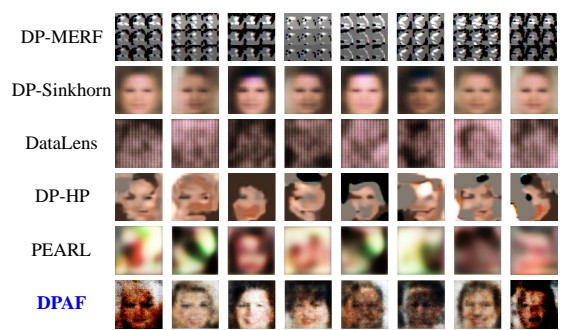

Figure 3: Visual results for $64 \times 64$ CelebA-Gender with $\varepsilon = 10$. The left (right) four columns are females (males).

### 4.2.3 HIGH RESOLUTION $128 \times 128$ IMAGE SYNTHESIS

To further demonstrate the advantage of DPAF in synthesizing high-utility and high-dimensional images, we conducted experiments on FFHQ-Gender, and the visual results are shown in Figure 8 in Appendix A.9, where the images synthesized by PEARL look like pure noise, while the images synthesized by DPAF still preserve facial features. Table 3 reports the predicting accuracy of FFHQ-Gender. In Section 4.2.1, we claim that the generative ability of DPAF increases with image size. At first glance, we can see from Table 3 that the accuracy gets worse compared to CelebA-Gender in Table 1. However, this can be explained as follows. First, CelebA-Gender and FFHQ-Gender are two different datasets with different distributions. The direct comparison between the accuracy of CelebA-Gender and FFHQ-Gender remains doubtful. Second, synthesizing $128 \times 128$ images has reached the limit of conventional GANs without using modern techniques such as residual blocks (He et al., 2016). Synthesizing higher resolution images requires much more sophisticated generative models (e.g., PGGAN (Karras et al., 2018) and DDPM (Ho et al., 2020)). Applying the techniques in DPAF to more sophisticated generative models remains unexplored, but would be our future research direction.

### 4.2.4 PRIVACY BUDGET ALLOCATION W/O TRANSFER LEARNING

Since we allocate a very limited privacy budget to conv1 in our canonical implementation, a natural question is whether transfer learning is necessary. In other words, if conv1 cannot learn effectively from the data, a reasonable design choice is to abandon transfer learning and invest the privacy

budget in DPAGG. Below, we examine the predicting accuracy under three strategies for allocating the privacy budget in the absence of transfer learning.

**Random Parameters for conv1.** We considered random weights of conv1; i.e., all weights in conv1 are sampled uniformly at random from a zero-mean Gaussian distribution with a standard deviation of 0.02 and are never updated. The results are shown in Table 4, where the notation $(\times, x_3)$ means that the privacy budget $\varepsilon_3 = x_3$ is allocated to

|  | $\varepsilon$ | DPAF | $(\times, 0.5)$ | $(\times, 0.2)$ |
|---|---|---|---|---|
| CelebA-Gender | 1 | $\mathbf{0.802} \pm 0.018$ | $0.752 \pm 0.045$ | $0.774 \pm 0.012$ |
| | 10 | $\mathbf{0.826} \pm 0.010$ | $0.793 \pm 0.024$ | $0.700 \pm 0.093$ |
| CelebA-Hair | 1 | $\mathbf{0.675} \pm 0.013$ | $0.635 \pm 0.035$ | $0.667 \pm 0.029$ |
| | 10 | $0.671 \pm 0.014$ | $\mathbf{0.681} \pm 0.015$ | $0.670 \pm 0.016$ |

Table 4: The accuracy of random parameters for conv1.

DPAGG, while the rest of the budget is allocated to conv2*. One can see that the accuracy of $(\times, 0.5)$ and $(\times, 0.2)$ is only slightly lower than the canonical accuracy. Since conv1 is supposed to learn low-level features, even if conv1 uses random features, the learning of conv2*, conv3*, and FC* can be adapted to random conv1 and perform well. However, according to our empirical experience, we still spent a very limited privacy budget on conv1 to avoid undesirable cases where some feature maps happen to contain only zero or near-zero values, rendering such feature maps useless. Higher variances of $(\times, 0.5)$ and $(\times, 0.2)$ also justify the above design choice.

**Updating conv1 During Training of** $D$**.** Here we did not perform transfer learning, but still used DPSGD($\varepsilon_1$) to update conv1 during training of $D$. Table 5 shows the results, where the canonical implementation outperforms the other configurations. There are two reasons for this. First, while the update of conv1 is done during the training of

|  | $\varepsilon$ | DPAF | $(0.1, \times, 0.1)$ | $(33\%, 34\%, 33\%)$ |
|---|---|---|---|---|
| CelebA-Gender | 1 | $\mathbf{0.802} \pm 0.018$ | $0.594 \pm 0.112$ | $0.727 \pm 0.143$ |
| | 10 | $\mathbf{0.826} \pm 0.010$ | $0.747 \pm 0.100$ | $0.817 \pm 0.024$ |
| CelebA-Hair | 1 | $\mathbf{0.675} \pm 0.013$ | $0.424 \pm 0.079$ | $0.642 \pm 0.038$ |
| | 10 | $0.671 \pm 0.014$ | $0.599 \pm 0.116$ | $\mathbf{0.685} \pm 0.018$ |

Table 5: Acc. of updating conv1 during training of $D$.

$C$, since conv2, conv3, and FC in $C$ are updated by SGD, conv1 is more informative. While conv1 is updated during training of $D$, conv1 is less informative due to noise accumulation. Second, in the absence of transfer learning, more layers (parameters) need to be updated during the training of $D$, which is more difficult to train well from an adversarial learning perspective. Note that $(0.1, \times, 0.1)$ in the second column of Table 5 means that we updated conv1 by DPSGD($\varepsilon_1$) with $\varepsilon_1 = 0.1$ in the training of $D$, although the same $(0.1, \times, 0.1)$ has a different interpretation in the context of using transfer learning, as shown in Section 4.1.

**Joint Updating conv1 and conv2\* During Training of** $D$**.** In this case, conv1 and conv2* are considered together and are updated together by the DPSGD. We see (conv1, conv2*) as a larger component. Compared to the individual conv1 and conv2*, gradient clipping in DPSGD may cause more information loss, and DPSGD will cause more damage to the gradient structure. The results in Table 6 support the above arguments.

Furthermore, by comparing the $(0.1, \times, 0.1)$ column in Table 5 and the $(\times, 0.1)$ column in Table 6 (due to their similar settings), we can see that the accuracy of the former is consistently lower than that of the latter. Unlike Table 6, conv1 and con2* are treated separately in Table 5, and thus suffer from privacy budget splitting, resulting in worse accuracy.

|  | $\varepsilon$ | DPAF | $(50\%, 50\%)$ | $(\times, 0.1)$ |
|---|---|---|---|---|
| CelebA-Gender | 1 | $\mathbf{0.802} \pm 0.018$ | $0.768 \pm 0.036$ | $0.748 \pm 0.072$ |
| | 10 | $\mathbf{0.826} \pm 0.010$ | $0.759 \pm 0.030$ | $0.790 \pm 0.020$ |
| CelebA-Hair | 1 | $\mathbf{0.675} \pm 0.013$ | $0.661 \pm 0.034$ | $0.643 \pm 0.028$ |
| | 10 | $\mathbf{0.671} \pm 0.014$ | $0.648 \pm 0.038$ | $0.660 \pm 0.018$ |

Table 6: The classification accuracy of joint updating conv1 and conv2* during training of $D$.

### 4.2.5 PRIVACY BUDGET ALLOCATION W/ TRANSFER LEARNING

Given the canonical use of DPAF (i.e., DPAF with transfer learning), we aim to examine the impact of different budget allocations on accuracy.

A general guideline for allocating privacy budgets is that the earlier (latter) layers should earn more (fewer) budgets. The rationale is that the earlier layers learn the low-level features and the latter layers will be adapted to the low-level features. Once the earlier layers have only a limited budget and the parameters fluctuate, the latter layers can hardly be adapted to the fast change of earlier layers and can hardly learn informative parameters. However, during the training of $D$, conv1 is frozen and does not need to be updated. In addition, as mentioned in Section 4.2.4, $\varepsilon_1$ can be a small value. The gradient vector may have many small or even nearly zero values, which can easily be affected by the DP noise. On the other hand, the aggregated vector output by DPAGG is designed to have larger values for better robustness against the DP noise. Thus, a reasonable choice is to have $\varepsilon_2 > \varepsilon_3$. The above arguments can be confirmed empirically because we can see from Table 7 that the canonical setting $(0.1, \times, 0.1)$ that follows the above discussion outperforms the other settings.

| | $\varepsilon$ | DPAF $(0.1, \times, 0.1)$ | (20%, 20%, 60%) | (20%, 60%, 20%) | (20%, 40%, 40%) | (30%, 20%, 50%) | (30%, 50%, 20%) |
|---|---|---|---|---|---|---|---|
| CelebA- | 1 | **0.802** $\pm$ 0.018 | 0.741 $\pm$ 0.074 | 0.801 $\pm$ 0.035 | 0.793 $\pm$ 0.030 | 0.771 $\pm$ 0.035 | 0.795 $\pm$ 0.033 |
| Gender | 10 | **0.826** $\pm$ 0.010 | 0.787 $\pm$ 0.018 | 0.820 $\pm$ 0.020 | 0.790 $\pm$ 0.017 | 0.767 $\pm$ 0.040 | 0.813 $\pm$ 0.013 |
| CelebA- | 1 | 0.675 $\pm$ 0.013 | 0.570 $\pm$ 0.144 | 0.663 $\pm$ 0.031 | 0.657 $\pm$ 0.023 | 0.643 $\pm$ 0.046 | 0.666 $\pm$ 0.024 |
| Hair | 10 | **0.671** $\pm$ 0.014 | 0.639 $\pm$ 0.031 | 0.619 $\pm$ 0.038 | 0.659 $\pm$ 0.016 | 0.598 $\pm$ 0.050 | 0.577 $\pm$ 0.094 |

| | $\varepsilon$ | (30%, 40%, 30%) | (30%, 30%, 40%) | (40%, 30%, 30%) | (40%, 20%, 40%) | (40%, 40%, 20%) |
|---|---|---|---|---|---|---|
| CelebA- | 1 | 0.791 $\pm$ 0.012 | 0.704 $\pm$ 0.164 | 0.745 $\pm$ 0.102 | 0.759 $\pm$ 0.025 | 0.763 $\pm$ 0.038 |
| Gender | 10 | 0.799 $\pm$ 0.022 | 0.799 $\pm$ 0.023 | 0.797 $\pm$ 0.015 | 0.782 $\pm$ 0.032 | 0.798 $\pm$ 0.022 |
| CelebA- | 1 | **0.677** $\pm$ 0.011 | 0.653 $\pm$ 0.024 | 0.646 $\pm$ 0.039 | 0.454 $\pm$ 0.150 | 0.641 $\pm$ 0.064 |
| Hair | 10 | 0.610 $\pm$ 0.018 | 0.643 $\pm$ 0.012 | 0.628 $\pm$ 0.011 | 0.634 $\pm$ 0.036 | 0.645 $\pm$ 0.021 |

Table 7: The classification accuracy of different privacy budget allocations with transfer learning.

| | $\varepsilon$ | $\mu = 2$ | $\mu = 4$ | DPAF ($\mu = 8$) | $\mu = 10$ | $\mu = 20$ |
|---|---|---|---|---|---|---|
| CelebA- | 1 | 0.775 $\pm$ 0.037 | 0.688 $\pm$ 0.070 | **0.802** $\pm$ 0.018 | 0.772 $\pm$ 0.036 | 0.773 $\pm$ 0.030 |
| Gender | 10 | 0.665 $\pm$ 0.162 | 0.819 $\pm$ 0.033 | **0.826** $\pm$ 0.010 | 0.745 $\pm$ 0.010 | 0.735 $\pm$ 0.110 |
| CelebA- | 1 | 0.578 $\pm$ 0.094 | 0.642 $\pm$ 0.038 | **0.675** $\pm$ 0.013 | 0.656 $\pm$ 0.017 | 0.648 $\pm$ 0.011 |
| Hair | 10 | 0.670 $\pm$ 0.027 | 0.666 $\pm$ 0.041 | **0.671** $\pm$ 0.014 | 0.670 $\pm$ 0.015 | 0.668 $\pm$ 0.020 |

Table 8: The classification accuracy of different asymmetry multipliers $\mu$'s.

### 4.2.6 THE IMPACT OF ASYMMETRY MULTIPLIER $\mu$.

A larger $\mu$ implies a much larger budget for updating conv2*, given $\varepsilon_3$ for DPAGG. Obviously, the increased $\mu$ raises accuracy because conv2* virtually has more budget. Moreover, Table 8 supports our claim in Appendix A.5 that $\mu$ cannot be arbitrarily increased. The reason is that the increased $\mu$ also leads to a less frequent update of conv2*, which may, in turn, degrade the utility.

### 4.2.7 EXTRA EXPERIMENTS

Appendix A.10~A.13 includes extra experiments results. For example, Appendix A.10 reports the optimal number of layers for conv1, conv2*, conv3*. Appendix A.11 examines three general techniques for improving the model utility, pre-training the model with public data (De et al., 2022; Zhang et al., 2018; Tramèr & Boneh, 2021), gradient compression (Lin et al., 2018; Wang et al., 2021), and tempered sigmoid activation (Papernot et al., 2021), to see whether they provide similar benefits to DPAF. A concurrent work, Private-GANs (Bie et al., 2023), is conceptually similar to but can be seen as an oversimplified version of it. Appendix A.12 shows that DPAF outperforms Private-GANs. Furthermore, in addition to a formal privacy proof in Appendix A.6, we provide additional empirical evidence in Appendix A.13 that the DPAF-synthesized dataset can resist MIA.

## 5 CONCLUSION

Overall, we propose a novel and effective DPGAN, DPAF, which can synthesize high-dimensional image data. Fundamentally different from the prior works, DPAF is featured by the DP feature aggregation in the forward phase, which significantly improves the robustness against noise. In addition, we propose a novel asymmetric training strategy, which determines an ideal batch size. We formally prove the privacy of DPAF. Extensive experiments demonstrate superior performance compared to the previous state-of-the-art methods.

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
