## A    APPENDIX

### A.1    DPSGD

Given a training set $\{(x_i, y_i)\}_{i=1}^{N}$, the update of the ordinary stochastic gradient descent (SGD) is formulated as $w^{t+1} = w^{(t)} - \eta_t \frac{1}{B} \sum_{i \in \mathcal{B}_t} \nabla \mathcal{L}(w^{(t)}, x_i, y_i)$, where $\mathcal{L}(w, x, y)$ is the loss function with the model parameter $w$, input sample $x$, and label $y$, and $\mathcal{B}_t$ is the set of samples at iteration $t$ with $|\mathcal{B}_t| = B$. As the gradient has an unbounded sensitivity, one has to clip the gradient to ensure a bounded DP noise magnitude. Formally, the update of DPSGD can be formulated below.

$$w^{t+1} = w^{(t)} - \eta_t \left\{ \frac{1}{B} \sum_{i \in \mathcal{B}_t} \text{clip}_u \left( \nabla \mathcal{L}(w^{(t)}, x_i, y_i) \right) + \frac{\sigma u}{B} \xi \right\}, \tag{3}$$

where $\eta_t$ is the learning rate, $\xi$ is sampled from the zero-mean Gaussian distribution, $\sigma$ specifies the standard deviation of the added noise, and $\text{clip}_u$ is defined as $\text{clip}_u(v) = \min\{1, \frac{u}{||v||_2}\} \cdot v$ with $u$ as a manually configured clipping threshold. Usually, $u$ is used as the $\ell_2$-sensitivity for SGD.

### A.2    PSEUDOCODE OF DPAF

The pseudocode of DPAF is shown in Algorithm 1.

---

**Algorithm 1:** Training of DPAF

1 **Notation**: number of batches $\mathbb{B}$, mean square error loss function $\mathcal{L}_{\text{MSE}}$, binary cross-entropy loss functions $\mathcal{L}_{\text{BCE}}, \mathcal{L}'_{\text{BCE}}, \mathcal{L}''_{\text{BCE}}$, asymmetry multiplier $\mu$, number of critic iterations per generator iteration $n_{\text{critic}}$

```
/* the for-loop below trains the classifier C                        */
```
2 **for** $i = 1$ *to* $\mathbb{B}$ **do**
3    compute $\mathcal{L}_{\text{MSE}}$ over the $i$-th batch
4    perform SGD for updating conv2, conv3, and FC and then perform DPSGD($\varepsilon_1$) for updating conv1 in one round of backpropagation

5 conv1* $\leftarrow$ conv1                                  `// conv1* and conv1 share parameters`
6 **for** $i = 1$ *to* $\mathbb{B}$ **do**
```
   /* computing L_BCE on D with DPAGG(ε₃), L'_BCE on D that replaces
      DPAGG(ε₃) by AGG, and L''_BCE on D without DPAGG(ε₃), respectively */
```
7    compute $\mathcal{L}_{\text{BCE}}$ over the $i$-th batch
```
   /* the code below asymmetrically trains D                         */
```
8    **if** $i\%\mu = 0$ **then**
9        compute $\mathcal{L}'_{\text{BCE}}$ over the $[i - \mu + 1, i]$-th batches
10       SGD for updating conv3* and FC* by $\mathcal{L}_{\text{BCE}}$
11       DPSGD($\varepsilon_2$) for updating conv2* by $\mathcal{L}'_{\text{BCE}}$
12   **else**
13       SGD for updating conv3* and FC* by $\mathcal{L}_{\text{BCE}}$
```
   /* the code below trains G                                        */
```
14   **if** $i\%n_{critic} = 0$ **then**
15       compute $\mathcal{L}''_{\text{BCE}}$ over each sample from the $i$-th batch
16       SGD for update of $G$

---

### A.3    NOTATION TABLE

The notation table summarizing the frequently used notations can be found in Table 9.

### A.4    DESIGN IDEA OF DPAF

Unlike previous DPGANs, DPAF has a fundamentally different design where DP feature aggregation is performed in the forward phase. The aggregated feature makes the image features more

| Symbol | Description |
|---|---|
| $\mathcal{D}, \mathcal{D}'$ | The neighboring data |
| $C$ | The classifier in DAF before transfer learning |
| $G$ | The generator in DAF after transfer learning |
| $D$ | The discriminator in DAF after transfer learning |
| $\mu$ | Asymmetry multiplier |
| $n_{\text{critic}}$ | Number of critic iterations per generator iteration |
| $\epsilon$ | The privacy loss |
| $\delta$ | The probability of violating DP |
| $\sigma^2$ | The variance of Gaussian distribution |
| $M_f$ | The feature extractor (FE) |
| $M_c$ | The label predictor |
| $\theta_f$ | The parameters of the FE |
| $\theta_c$ | The parameters of the label predictor |
| $u$ | The clipping threshold (sensitivity of DPSGD) |
| $\text{clip}_u$ | Gradient clipping function with threshold $u$ |
| $w$ | The model parameter |
| $p$ | The size of feature map is $p \times p$ |
| $m$ | The number of feature maps |
| $\mathbb{B}$ | The number of batches |
| IN | The instance normalization |
| SIN | The simplified instance normalization |
| $\mu_{i_1 i_2}$ | The mean of feature map $X_{i_1 i_2}$ |
| $\sigma^2_{i_1 i_2}$ | The variance of feature map $X_{i_1 i_2}$ |
| $H$ | The height of the feature map |
| $W$ | The width of the feature map |
| $x_{i_1 i_2 i_3 i_4}$ | The element of feature map $X_{i_1 i_2}$ |
| $\widehat{x_{i_1 i_2 i_3 i_4}}$ | The new value of $x_{i_1 i_2 i_3 i_4}$ after SIN |
| $\alpha$ | The order in Rényi DP |
| $D_\alpha$ | The Rényi divergence of order $\alpha$ |
| $G_\sigma$ | The Gaussian mechanism with variance $\sigma^2$ |
| $\gamma$ | The subsampling rate |

Table 9: Notation Table

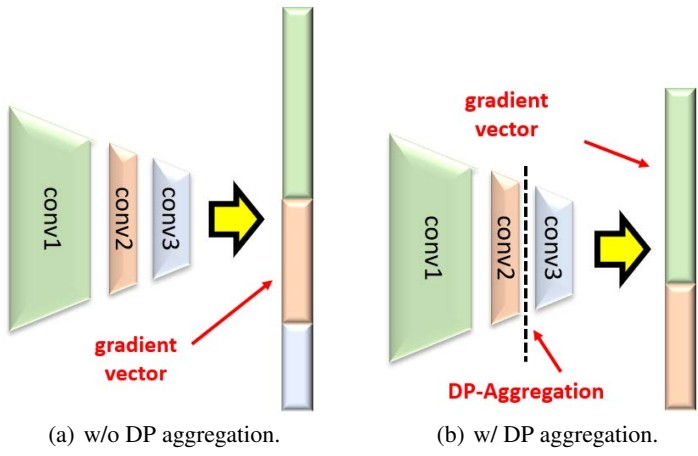

(a) w/o DP aggregation.  (b) w/ DP aggregation.

Figure 4: The illustration of the impact of DP feature aggregation on the size of the gradient vector.

robust against the DP noise. On the other hand, the DP feature aggregation in the forward phase implies a shortened gradient vector, resulting in a significant reduction of information loss in gradient clipping. DPAF is also characterized by the use of a simplified instance normalization that preserves fine-grained features and reduces $\ell_2$ sensitivity. Overall, the five advantages of performing DP feature aggregation in the forward phase are summarized below.

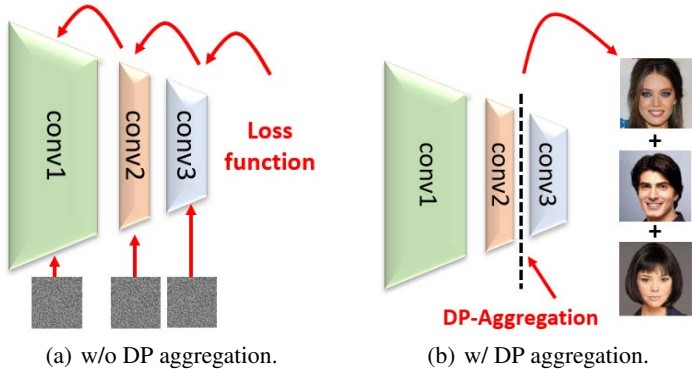

(a) w/o DP aggregation.       (b) w/ DP aggregation.

Figure 5: The impact of DP feature aggregation on the gradient structure preservation during the backpropagation.

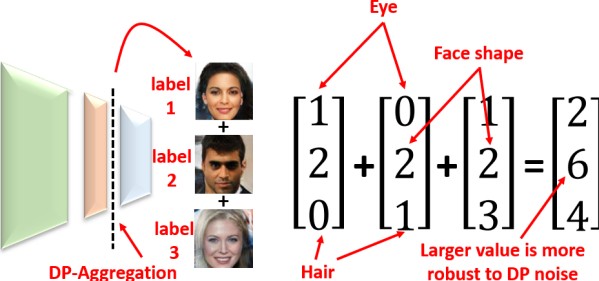

Figure 6: The illustration of the better robustness against the DP noise after the feature aggregation.

**Reduction of Information Loss in Gradient Clipping.** The first advantage is the dimensionality reduction of the gradient vector, as illustrated in Figure 4. More specifically, gradient clipping in DPSGD inevitably leads to information loss. However, gradient clipping has less impact on shorter gradient vectors, resulting in less information loss. As shown in Figure 4(a), the gradient vector to be clipped will be longer if DP feature aggregation is not used. On the contrary, as shown in Figure 4(b), conv3 has been privatized after DP feature aggregation due to the postprocessing property of DP, and can be updated by SGD. As a result, since only conv1 and conv2 need to be updated by DPSGD, the information loss due to gradient clipping can be mitigated.

**Better Preserving Gradient Structure.** During backpropagation, DPSGD applies noise to the weights in a layer-by-layer manner, as shown in Figure 5(a), which makes training more difficult because such an updating process destroys the inherent structure of the gradient vector. Thus, the second advantage is to better preserve the gradient structure. This is due to the fact that the aggregated DP vector is a vector of aggregated noisy image features. As shown in Figure 5(b), since the aggregated features still have the inherent semantics, the corresponding noisy version remains meaningful. On the other hand, as also shown in Figure 5(b), conv3 can be updated by SGD instead of DPSGD, which better preserves the inherent gradient structure. In this case, only layers (e.g. conv2) need to be updated by DPSGD, and as a result, only a small fraction of the parameter structure will be affected by DPSGD.

**Better Robustness against DP Noise.** The aggregation of the features from samples makes the aggregated feature more robust to the DP noise, because the DP noise is added after the feature value summation. This is illustrated in Figure 6, where each individual feature is relatively susceptible to the DP noise, but the aggregated one has the larger values and, as a result, better robustness.

**Better Preserving Image Features.** From Figure 6, we know that the aggregated feature vector, which is robust to the DP noise, helps in synthesizing realistic samples (because it is backpropagated to update $G$), but such synthetic samples may be irrelevant for a particular label. In fact, the feature

values should have similar numerical ranges, otherwise, $D$ will only pay attention to the features with large values and ignore the features with small values. As a consequence, $G$ cannot be updated well. We propose to use a simplified instance normalization (SIN) to ensure that fine-grained features can be learned. Specifically, SIN is applied to each feature map individually. Then, the feature vector (concatenated normalized feature maps) undergoes aggregation. For example, this helps in synthesizing faces with consistent gender in the conditional generation of faces. In other words, in general, without SIN, due to the imbalance of feature values, some feature values will be devoured by the others, resulting in the disappearance of certain important feature values that are related to the specific class.

**Low Global Sensitivity.** The fifth advantage is the low $\ell_2$-sensitivity of the SIN-and-aggregation operation. More specifically, as mentioned above, the feature maps need to be normalized and then concatenated as the feature vector before the aggregation. We find that the $\ell_2$-sensitivity of such an SIN-and-aggregation operation can be calculated as a relatively small and controllable value $\sqrt{mp}$, where $m$ is the number of feature maps and the size of the feature map is $p \times p$. Properly setting $\sqrt{mp}$ effectively reduces the noise magnitude, thereby raising the utility (see Appendix A.5).

## A.5    DETAILED CONFIGURATION OF DPAF

Here, we discuss the rationale behind the design of DPAF.

**Why Not Eliminate conv2\*.** Consider the case where all of the layers before the DP feature aggregation belong to conv1\*. The number of learnable parameters in $D$ will be much smaller (i.e., only conv3\* and FC\*); i.e., no conv2\* exists. Such a setting hurts the training of GANs. This can be attributed to the fact that one knows from the GAN literature that if $G$ ($D$) is much stronger than $D$ ($G$), the training of GANs will likely fail to converge. In addition, conv3\* and FC\* might have fewer parameters compared to conv2\*, depending on different model structures. It is difficult to well train $D$ under this circumstance. Thus, keeping certain layers as conv2\* is beneficial for adversarial learning.

**Why Not More Layers for conv2\*.** As more learnable parameters in $D$ may help the training of GANs, why conv2\* does not have more layers? This can be explained as follows. If conv2\* has more layers (parameters), because conv2\* is updated through DPSGD, gradient clipping will lead to more information loss, flattening the feature values. In addition, if conv2\* has more layers (parameters), because conv1 and conv2\* both are trained by DPSGD, the output of (conv1, conv2\*) will be too noisy, hindering the utility.

**Why Not More Layers for conv3\*.** A question that may arise is why conv3\* does not have more layers. As the total number of layers is fixed given an input image, if conv3\* has more layers, then either conv1 or conv2\* (or both) will be shrunk. Thus, DPAGG is closer to low-level features. In such a case, $D$ cannot have meaningful learning from the aggregation of level-level features.

**Choice of cGAN.** The DPAF is designed to support conditional generation. Thus, one needs to consider a cGAN in DPAF. Compared to GANs, $G$ and $D$ of cGANs need to consider the class label to ensure both the indistinguishability between the real and synthetic samples and the consistency between the input label and the label of synthetic samples. In general, there are two straightforward solutions for label injection. First, the class label is added as part of the input vector in such a case. If we feed labels to the input layer, the labels will be diluted in the forward phase and have a weak signal only. Second, $D$ is designed with two loss functions; one for the ordinary GAN loss and another for the class label. A representative of such a design choice is AC-GAN (Odena et al., 2017), which outputs labels as part of the loss function. Nevertheless, due to access to the label, such a design leads to a privacy budget splitting and therefore suffers from utility degradation.

In our design, DPAF follows the architecture of cDCGAN (Radford et al., 2016). Inspired by (Perarnau et al., 2016) stating that the class label is better added to the first layer, we decide to use cDCGAN though there are no considerations of aggregation and DP in (Perarnau et al., 2016). In essence, cDCGAN feeds labels to the second layer by first computing the embedding (from scalar to vector) of labels, significantly strengthening the signal. A natural question that arises is why the class label is not added to the latter layers of $D$, given the class label in the latter layers may preserve

an even stronger signal. The drawback of doing so is that all the layers before the layer to which the class label is added can hardly learn anything, because the classifier can know the portion by looking at the label only.

**The Position of DPAGG.** DPAF heavily relies on DPAGG to raise the utility of synthetic samples. Thus, a natural question that arises is where the best position for DPAGG is. Without the loss of generality, DPAGG is placed to minimize the $\ell_2$-sensitivity $\Delta_{2,\text{AGG}}$ in Eq. (2). As $\Delta_{2,\text{AGG}} = \sqrt{mp}$, $\Delta_{2,\text{AGG}}$ is dependent on the position of DPAGG. Given the design of a conventional CNN, where the height/width of feature maps in the next convolutional layer is half of the ones in the current convolutional layer, if the input is a $\rho \times \rho$ $c$-channel image, $\Delta_{2,\text{AGG}}$ can be calculated as $\sqrt{c \prod_{i=1}^{a} k_i} \cdot \frac{\rho}{2^a}$, where $k_i$ is the number of filters in the $i$-th convolutional layer and DPAGG is placed behind the $a$-th convolutional layer. Define $R_j$ as $\left( \sqrt{c \prod_{i=1}^{j} k_i} \cdot \frac{\rho}{2^j} \right) / \left( \sqrt{c \prod_{i=1}^{j-1} k_i} \cdot \frac{\rho}{2^{j-1}} \right)$. We can easily derive $R_j = \sqrt{k_j}/2$. Thus, if $k_j \geq 4$, then $\Delta_{2,\text{AGG}}$ is monotone increasing from earlier to latter layers. As a consequence, from the $\ell_2$-sensitivity point of view, we conclude that the best position for DPAGG is between the first and second convolutional layers. Unfortunately, placing DPAGG in such a position does not lead to a decent utility in practice, because it completely destroys the structure of DPAF (e.g., the disappearance of conv1 and conv2*). Hence, we will empirically examine the other configurations in Section 4.

### A.6 PRIVACY ANALYSIS

At the beginning, we aim to prove that conv1 satisfies DP. We start from Theorem 1.

**Theorem 1.** *Define a model $M(\mathcal{D}) = M_2(\theta_2, M_1(\theta_1, \mathcal{D})) : \mathcal{D} \to R_2$, where $\mathcal{D}$ is the sensitive data, $\theta_1$ and $\theta_2$ are model parameters, $M_1(\theta_1, \mathcal{D}) : \mathcal{D} \to R_1$ is the first half of the model $M$, $M_2(\theta_2, R_1) : R_1 \to R_2$ is the second half of the model $M$, with $R_1$ and $R_2$ denoting the corresponding outputs of the layers. $M_1(\theta_1, \mathcal{D})$ satisfies DP if it is trained by DPSGD.*

*Proof.* Define $G(\theta_1, \theta_2, \mathcal{D}, \mathcal{L}(\theta_1, \theta_2, \mathcal{D}))$ as the gradient of the model $M$, where $\mathcal{L}$ is the loss function. $G(\theta_1, \theta_2, \mathcal{D}, \mathcal{L}(\theta_1, \theta_2, \mathcal{D}))$ can be rewritten as

$$G(\theta_1, \theta_2, \mathcal{D}, \mathcal{L}(\theta_1, \theta_2, \mathcal{D})) \tag{4}$$
$$= [g(\theta_2, \mathcal{L}(\theta_1, \theta_2, \mathcal{D})), \tilde{g}(\theta_1, g(\theta_2, \mathcal{L}(\theta_1, \theta_2, \mathcal{D})))], \tag{5}$$

where $g(\theta_2, \mathcal{L}(\theta_1, \theta_2, \mathcal{D}))$ can be seen as the gradient of $M_2$ and $\tilde{g}(\theta_1, \mathcal{L}(\theta_1, \theta_2, \mathcal{D}))$ can be seen as the gradient of $M_1$. The updating rule is shown below.

$$\theta_2 \leftarrow \theta_2 + g(\theta_2, \mathcal{L}(\theta_1, \theta_2, \mathcal{D})) \tag{6}$$
$$\theta_1 \leftarrow \theta_1 + \tilde{g}(\theta_1, g(\theta_2, \mathcal{L}(\theta_1, \theta_2, \mathcal{D}))) \tag{7}$$

To simplify the notations, we use $\mathbb{D}$ to denote the dependency of the sensitive data. In this case, Eq. (4) can be rewritten as

$$G(\theta_1, \theta_2, \mathcal{D}, \mathcal{L}(\theta_1, \theta_2, \mathcal{D}))$$
$$= G(\theta_1, \theta_2, \mathbb{D}) \tag{8}$$
$$= [g(\theta_2, \mathbb{D}), \tilde{g}(\theta_1, g(\theta_2, \mathbb{D}))]. \tag{9}$$

As $\tilde{g}(\theta_1, \mathbb{D})$ is trained by DPSGD, one can ensures that $\tilde{g}(\theta_1, \mathbb{D})$ satisfies DP. As $M_1(\theta_1, \mathcal{D})$ is initialized randomly and is updated by $\tilde{g}(\theta_1, \mathbb{D})$, one can ensures that $M_1(\theta_1, \mathcal{D})$ satisfies DP. $\square$

With Theorem 1, we can easily conclude that conv1 in DPAF satisfies DP via Corollary 1.

**Corollary 1.** *The conv1 in DPAF satisfies DP.*

*Proof.* The classifier $C$ in DPAF before the transfer learning is an instantiation of the model $M$ in Theorem 1. In this case, conv1 can be seen as $M_1$, and conv2, conv3, and FC are seen as $M_2$ in Theorem 1. As shown in Line 4 of Algorithm 1, we perform SGD for updating conv2, conv2, and FC, and perform DPSGD($\varepsilon_1$) for updating conv1. Hence, conv1 satisfies DP. $\square$

Next, we are aimed to prove that DPAF satisfies DP. We rely on Rényi DP (RDP) (Mironov, 2017) for our privacy analysis. Compared to the ordinary DP in Definition 1, RDP is a variant of DP with a tighter bound of privacy loss.

**Definition 2.** *A randomized algorithm $\mathcal{M}$ is $(\alpha, \epsilon(\alpha))$-RDP with $\alpha > 1$ if for any neighboring datasets $\mathcal{D}$ and $\mathcal{D}'$,*

$$D_\alpha(\mathcal{M}(\mathcal{D})||\mathcal{M}(\mathcal{D}')) = \frac{1}{\alpha - 1} \log \mathbb{E}_{x \sim \mathcal{M}(\mathcal{D}')} \left[ \left( \frac{Pr[(\mathcal{D}) = x]}{Pr[(\mathcal{D}') = x]} \right)^{\alpha - 1} \right] \leq \epsilon(\alpha), \tag{10}$$

*where $D_\alpha$ is the Rényi divergence of order $\alpha$.*

Before proving our main result, we describe some necessary properties of RDP in Theorems 2∼5.

**Theorem 2** (Gaussian Mechanism on RDP (Mironov, 2017; Wang et al., 2021)). *If a function $f$ has $\ell_2$-sensitivity $u$, then $G_\sigma \circ f$ obeys $(\alpha, \varepsilon(\alpha))$-RDP, where $\varepsilon(\alpha) = \alpha u^2/(2\sigma^2)$ and $G_\sigma$ is the Gaussian mechanism defined in Section 2.*

**Theorem 3** (Sequential Composition on RDP (Mironov, 2017)). *If the mechanism $\mathcal{M}_1$ satisfies $(\alpha, \epsilon_1)$-RDP and the mechanism $\mathcal{M}_2$ satisfies $(\alpha, \epsilon_2)$-RDP, then $\mathcal{M}_2 \circ \mathcal{M}_1$ satisfies $(\alpha, \epsilon_1 + \epsilon_2)$-RDP.*

**Theorem 4** (Privacy Amplification by Subsampling (Wang et al., 2019)). *Let $\mathcal{M} \circ$ subsample be a randomized mechanism that first performs the subsampling without replacement with the subsampling rate $\gamma$ on the dataset $X$ and then takes as an input from the subsampled dataset $X^\gamma$. For all integers $\alpha \geq 2$, if $\mathcal{M}$ obeys $(\alpha, \epsilon(\alpha))$-RDP through Gaussian mechanism, then $\mathcal{M} \circ$ subsample satisfies $(\alpha, \epsilon'(\alpha))$-RDP, where*

$$\epsilon'(\alpha) \leq \frac{1}{(\alpha - 1)} \log(1 + \gamma^2 \binom{\alpha}{2} \min\{4(e^{\epsilon(2)} - 1), e^{\epsilon(2)} \min\{2,$$

$$(e^{\epsilon(\infty)} - 1)^2\}\} + \sum_{j=3}^{\alpha} \gamma^j \binom{\alpha}{j} e^{(j-1)\epsilon(j)} \min\{2, (e^{\epsilon(\infty)} - 1)^j\}), \tag{11}$$

*and $\varepsilon(\alpha) = \alpha u^2/(2\sigma^2)$ with $u$ as the sensitivity.*

In the following, we use $\epsilon'(\alpha, \gamma, u)$ to indicate $\epsilon'(\alpha)$ with the subsampling rate $\gamma$ and sensitivity $u$.

**Theorem 5** (From RDP to DP (Mironov, 2017)). *If a mechanism $\mathcal{M}$ is $(\alpha, \epsilon(\alpha))$-RDP, $\mathcal{M}$ is $(\epsilon(\alpha) + \frac{\log 1/\delta}{\alpha - 1}, \delta)$-DP for any $\delta \in (0, 1)$.*

From Corollary 1, we know that if conv1 in $C$ is trained by DPSGD and conv2, conv3, and FC are discarded, using conv1 in $C$ as conv1 in $D$ does not leak privacy. Hence, based on the above result, Theorem 6 shows the DP of DPAF.

**Theorem 6.** *DPAF guarantees $(T_1\epsilon'(\alpha, \gamma_1, u_1) + T_2\epsilon'(\alpha, \gamma_2, u_2) + T_3\epsilon'(\alpha, \gamma_3, \sqrt{m}p) + \frac{\log \frac{1}{\delta}}{\alpha - 1}, \delta)$-DP for all $\alpha \geq 2$ and $\delta \in (0, 1)$.*

*Proof.* For $C$, our goal is to ensure that the update of conv1 is satisfied with $(\alpha, \epsilon(\alpha))$-RDP for each iteration. Note that because conv2, conv3, and FC will be discarded after the training, they do not need a DP guarantee. Let the total number of iterations for training $C$ be $T_1$. Then, the DPSGD (through the Gaussian mechanism in Theorem 2) on conv1 is $(\alpha, T_1\epsilon(\alpha))$-RDP according to Theorem 3. However, it can be re-estimated as $(\alpha, T_1\epsilon'(\alpha, \gamma_1, u_1))$-RDP with subsampling rate $\gamma_1$ according to Theorem 4.

For $D$, since conv1's parameters are frozen during the training of $D$, the output of conv1 in $D$ is satisfied with $(\alpha, T_1\epsilon'(\alpha, \gamma_1, u_1))$-RDP, because the update of conv1 in $C$ has been proven to be RDP. Let the total number of iterations for training conv2* be $T_2$. Then, the DPSGD (through the Gaussian mechanism in Theorem 2) on conv2* is $(\alpha, T_2\epsilon(\alpha))$-RDP according to and Theorem 3. Similarly, the update of conv2* is satisfied with $(\alpha, T_2\epsilon'(\alpha, \gamma_2, u_2))$-RDP with subsampling rate $\gamma_2$ according to Theorem 4. So far, the joint consideration of conv1 and conv2*, (conv1, conv2*), is satisfied with $(\alpha, T_1\epsilon'(\alpha, \gamma_1, u_1) + T_2\epsilon'(\alpha, \gamma_2, u_2))$-RDP guarantee according to Theorem 3. Unlike the cases of conv1 and conv2*, where noise is injected in the backward phase, the noise injection to AGG occurs in the forward phase. More specifically, we set a DPAGG to aggregate input data and add noise in the aggregated data. The sensitivity of AGG has been calculated as $\sqrt{m}p$ in Eq. (2).

Let the number of iterations for the training of $D$ be $T_3$. The DPAGG with the noise sampled from $N(0, mp^2\sigma^2)$ is satisfied with $(\alpha, T_3\epsilon'(\alpha, \gamma_3, \sqrt{mp}))$-RDP with subsampling ratio $\gamma_3$ according to Theorem 2 and Theorem 3. Thus, the joint consideration of conv1, conv2*, and DPAGG, (conv1, conv2*, DPAGG), will fulfill $(\alpha, T_1\epsilon'(\alpha, \gamma_1, u_1) + T_2\epsilon'(\alpha, \gamma_2, u_2) + T_3\epsilon'(\alpha, \gamma_3, \sqrt{mp}))$-RDP according to Theorem 3. Because the DPAGG has DP guarantee, the update of conv3* and FC* is satisfied with RDP by the postprocessing. Finally, the update of $G$ does not access the sensitive data and, as a result, is satisfied with RDP by the postprocessing. Overall, according to Theorem 5, DPAF is satisfied with $(T_1\epsilon'(\alpha, \gamma_1, u_1) + T_2\epsilon'(\alpha, \gamma_2, u_2) + T_3\epsilon'(\alpha, \gamma_3, \sqrt{mp}) + \frac{\log \frac{1}{\delta}}{\alpha - 1}, \delta)$-DP.

$\square$

### A.7 SOURCES OF OFFICIAL CODE FOR BASELINE METHODS

The official code of GS-WGAN, DP-MERF, DataLens, G-PATE, DP-Sinkhorn, DP-HP, and DPDC can be found at `https://github.com/DingfanChen/GS-WGAN`, `https://github.com/ParkLabML/DP-MERF`, `https://github.com/AI-secure/DataLens`, `https://github.com/AI-secure/G-PATE`, and `https://github.com/nv-tlabs/DP-Sinkhorn_code`, `https://github.com/ParkLabML/DP-HP`, and `https://openreview.net/attachment?id=H8XpqEkbua_&name=supplementary_material` respectively.

### A.8 THE ARCHITECTURE OF THE EVALUATED CLASSIFIER

Figure 7 shows the architecture of the classifier used in our experiments for the downstream classification task.

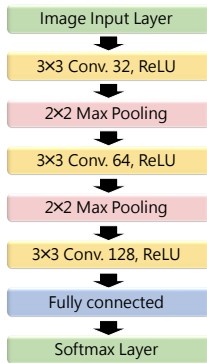

Figure 7: The architecture of the evaluated classifier.

### A.9 EXTRA EXPERIMENTAL RESULTS FOR VISUAL QUALITY

Visual results for $128 \times 128$ FFHQ images are shown in Figure 8, where the left (right) three columns are females (males).

### A.10 NUMBER OF LAYERS FOR CONV1, CONV2*, AND CONV3*

Our DPAF is configured to be C2-C1-$\times$ for MNIST/FMNIST, C2-C2-C1 for CelebA, and C3-C1-$\times$ for FFHQ, where the notation C$x_1$-C$x_2$-C$x_3$ means that the $D$ uses $x_1$ layers as conv1, $x_2$ layers as conv2*, and $x_3$ layers as conv3*. The notation $\times$ means that the corresponding layer does not exist. We always have two FC* layers.

Three datasets have different image sizes. Hence, the input layer for different datasets must have different dimensions, leading to different layer architectures. For example, according to the conventional CNN design (Section 3.1), a CNN for FFHQ may have up to five layers. However, the

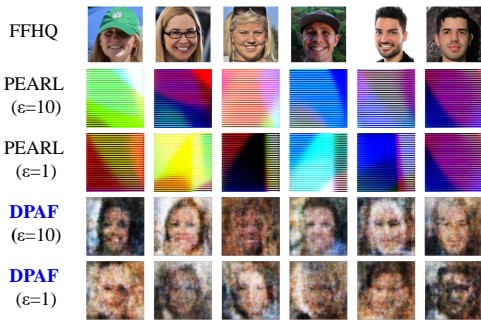

Figure 8: Visual results for $128 \times 128$ FFHQ images. The left (right) three columns are females (males).

arrangement of layers for conv1, conv2*, and conv3* is a hyperparameter. TABLE 11 shows the accuracy of different arrangements.

Using CelebA-Gender and CelebA-Hair as examples, we aim to know which layer configuration will result in better accuracy. As both CelebA-Gender and CelebA-Hair are $64 \times 64$, we know that there are at most five layers in total. Note that, in contrast to ordinary GANs, deliberately setting more layers in DPGANs may, in turn, hurt the training result (Bassily et al., 2014) because the lengthier gradient will lead to greater information loss, failing the convergence, according to our experience. There are too many configurations to exhaustively examine. Table 10 shows the only results of accuracy in the cases where conv1 and conv2* jointly occupy at most four layers. From Table 10, we know that C2-C1-$\times$, C2-C2-$\times$, C3-C1-$\times$, and C1-C3-$\times$ result in better accuracy. Thus, given the above results, we include the consideration of conv3* in Table 11, because Table 10 does not consider conv3*. The results in Table 11 support our design choice for the canonical implementation of C2-C2-C1 in DPAF because it outperforms the other settings.

|  | $\varepsilon$ | C1-C1-$\times$ | C1-C2-$\times$ | C2-C1-$\times$ | C1-C3-$\times$ | C2-C2-$\times$ | C3-C1-$\times$ |
|---|---|---|---|---|---|---|---|
| CelebA- | 1 | $0.629 \pm 0.040$ | $0.737 \pm 0.025$ | $\mathbf{0.824} \pm 0.025$ | $0.661 \pm 0.144$ | $0.805 \pm 0.021$ | $0.811 \pm 0.020$ |
| Gender | 10 | $0.720 \pm 0.045$ | $0.733 \pm 0.038$ | $0.762 \pm 0.079$ | $0.729 \pm 0.032$ | $\mathbf{0.786} \pm 0.018$ | $0.751 \pm 0.036$ |
| CelebA- | 1 | $0.423 \pm 0.089$ | $0.475 \pm 0.120$ | $0.643 \pm 0.014$ | $\mathbf{0.662} \pm 0.019$ | $0.519 \pm 0.095$ | $0.623 \pm 0.038$ |
| Hair | 10 | $0.565 \pm 0.019$ | $0.644 \pm 0.024$ | $0.639 \pm 0.039$ | $0.657 \pm 0.023$ | $\mathbf{0.683} \pm 0.022$ | $0.659 \pm 0.019$ |

Table 10: The classification accuracy of different layer architecture for conv1 and conv2*.

|  | $\varepsilon$ | DPAF (C2-C2-C1) | C1-C3-C1 | C2-C1-C1 | C2-C1-C2 | C3-C1-C1 |
|---|---|---|---|---|---|---|
| CelebA- | 1 | $\mathbf{0.802} \pm 0.018$ | $0.448 \pm 0.164$ | $0.673 \pm 0.099$ | $0.505 \pm 0.067$ | $0.800 \pm 0.017$ |
| Gender | 10 | $\mathbf{0.826} \pm 0.010$ | $0.727 \pm 0.039$ | $0.803 \pm 0.022$ | $0.514 \pm 0.091$ | $0.820 \pm 0.015$ |
| CelebA- | 1 | $\mathbf{0.675} \pm 0.013$ | $0.356 \pm 0.077$ | $0.540 \pm 0.036$ | $0.354 \pm 0.038$ | $0.669 \pm 0.018$ |
| Hair | 10 | $\mathbf{0.671} \pm 0.014$ | $0.368 \pm 0.074$ | $0.664 \pm 0.035$ | $0.352 \pm 0.055$ | $0.670 \pm 0.016$ |

Table 11: The classification accuracy of different layer architecture for conv1, conv2*, and conv3*.

## A.11 THE OTHER TECHNIQUES IN ENHANCING ACCURACY

Many techniques have been proposed to reduce the negative impact of DPSGD on model training. We examine three of them to see whether they provide similar benefits to DPAF.

**Pre-Training the Model with Public Data** The recent development of DP classifiers and DP-GANs has witnessed that extra data may help improve the performance of DP models (De et al., 2022; Zhang et al., 2018; Tramèr & Boneh, 2021). Here, we want to examine whether pre-training the model with public data helps DPAF raise its utility. Here, the common setting in Table 12 is that we follow DPAF to train $C$, perform transfer learning, and then train $G$ and $D$ on the CIFAR-10 dataset without considering DP. After that, DPAF is used to train DPGAN with the pre-trained $D$ as $D$ and the randomly initialized parameters as $G$. We additionally train DPGAN completely based

on the pre-trained parameters for both $G$ and $D$. One can see from Table 12 that the extra data still helps the utility of DPAF.

**The Impact of Gradient Compression.** Gradient compression (GC) (Lin et al., 2018) is originally proposed to reduce the communication cost in federated learning. The rationale behind gradient compression is that most of the values in the gradient contribute nearly no information on the update. Different from the original case, where GC works on the gradient averaged over the samples in a batch, the canonical implementation of DPAF adopts GC to keep only the top 90% values of per-sample gradients and then performs the averaging. However, we still want to examine whether GC can help DPSGD. The comparison between the DPAF column and "w/o GC" column in Table 13 still shows that DPSGD can benefit from GC because the information loss from gradient clipping can be mitigated. TOPAGG (Wang et al., 2021) is a modified DPSGD that works on compressed and quantized gradients. The GC in TOPAGG is configured to keep the top-$k$ values only[3]. Nevertheless, TOPAGG gains lower accuracy. This can be explained by considering the design of TOPAGG. In particular, the success of TOPAGG, in essence, relies on training a large number of teacher classifiers(Wang et al., 2021; Cao et al., 2021). As DPAF does not fit such a requirement, TOPAGG on DPAF does not perform well.

**Tempered Sigmoid Activation Function** Papernot et al. (2021) find that exploding activations cause the unclipped gradient magnitude to increase and therefore gradient clipping leads to more information loss. Thus, tempered sigmoid (TS) (Papernot et al., 2021), a family of activation functions, is proposed to replace the conventional activation functions in DPSGD. Table 14 shows the results, where hyperbolic tangent ($\mathtt{tanh}$), as a representative of TS, is used to replace the leaky ReLU in our canonical DPAF. In our test, $\mathtt{tanh}$ is used in DPAF, and we can see from Table 14 that it, in turn, leads to worse accuracy. This can be explained as follows. First, Papernot et al. (2021) conduct the experiments on DP classifiers only. Whether TS can raise the utility of DPGANs remains unknown. Second, a bounded activation (e.g., sigmoid and $\mathtt{tanh}$) easily causes gradient vanishing and therefore is rarely used in practice. On the contrary, the unbounded ones (e.g., ReLU and leaky ReLU) are more capable of avoiding gradient vanishing (Radford et al., 2016). conv1 and conv2* in DPAF are updated by DPSGD, but conv3* and FC* are updated by SGD. While TS is beneficial to DPSGD (for conv1 and conv2*) but harmful to SGD (conv2* and FC*). Overall, adopting $\mathtt{tanh}$ in DPAF slightly degrades utility.

|  | $\varepsilon$ | DPAF | Trans($D$) | Trans($G + D$) |
|---|---|---|---|---|
| CelebA-Gender | 1 | $0.802 \pm 0.018$ | $0.819 \pm 0.020$ | $\mathbf{0.820} \pm 0.027$ |
|  | 10 | $0.826 \pm 0.010$ | $0.830 \pm 0.018$ | $\mathbf{0.831} \pm 0.023$ |
| CelebA-Hair | 1 | $0.675 \pm 0.013$ | $0.663 \pm 0.038$ | $\mathbf{0.695} \pm 0.015$ |
|  | 10 | $0.671 \pm 0.014$ | $0.695 \pm 0.036$ | $\mathbf{0.703} \pm 0.015$ |

Table 12: The classification accuracy of DPAF with public data pre-training.

|  | $\varepsilon$ | DPAF | w/o GC | TOPAGG |
|---|---|---|---|---|
| CelebA-Gender | 1 | $\mathbf{0.802} \pm 0.018$ | $0.725 \pm 0.150$ | $0.549 \pm 0.075$ |
|  | 10 | $\mathbf{0.826} \pm 0.010$ | $0.818 \pm 0.022$ | $0.530 \pm 0.097$ |
| CelebA-Hair | 1 | $\mathbf{0.675} \pm 0.013$ | $0.673 \pm 0.016$ | $0.359 \pm 0.079$ |
|  | 10 | $0.671 \pm 0.014$ | $\mathbf{0.678} \pm 0.017$ | $0.375 \pm 0.056$ |

Table 13: The classification accuracy of DPAF with different compression strategies.

## A.12 COMPARISON TO PRIVATE-GANS

A concurrent work, Private-GANs (Bie et al., 2023), is conceptually similar to but can be seen as an oversimplified version of DPAF. Tables 15, 16, and 17 show the comparison between DPAF and Private-GANs. In particular, the results of DPAF in Table 15 are derived by optimizing both $\mu$ and $n_{\text{critic}}$ (called $n_D$ in their paper). As Private-GANs reported the results with

---

[3]For CelebA-Gender, $k = 200$ with $\varepsilon = 1$ and $k = 3000$ with $\varepsilon = 10$. For CelebA-Hair, $k = 150$ with $\varepsilon = 1$ and $k = 200$ with $\varepsilon = 10$.

|  | $\varepsilon$ | DPAF | w/ TS |
|---|---|---|---|
| CelebA-Gender | 1 | $\mathbf{0.802} \pm 0.018$ | $\mathbf{0.802} \pm 0.015$ |
|  | 10 | $\mathbf{0.826} \pm 0.010$ | $0.806 \pm 0.025$ |
| CelebA-Hair | 1 | $\mathbf{0.675} \pm 0.013$ | $0.594 \pm 0.031$ |
|  | 10 | $\mathbf{0.671} \pm 0.014$ | $0.620 \pm 0.043$ |

Table 14: The classification accuracy of DPAF with tempered sigmoid (TS) functions.

$n_{\text{critic}} = 1, 10, 50, 100, 200$, to be aligned with their results, Table 16 and 17 show the results of DPAF and Private-GANs for $n_{\text{critic}} = 1, 10, 50, 100, 200$. One can see from Tables 15, 16, and 17 that DPAF outperforms Private-GANs. This can be attributed to the fact that introducing $\mu$ in the design of DPAF as asymmetric training significantly reduces the downside from raising $n_{\text{critic}}$.

|  | $\varepsilon$ | NDPDC | PEARL | Private-GANs | DPAF |
|---|---|---|---|---|---|
| CelebA-Gender | 0.5 | 0.518 | 0.655 | 0.620 | **0.768** |
|  | 1 | 0.540 | 0.634 | 0.663 | **0.802** |
|  | 5 | 0.535 | 0.639 | 0.679 | **0.789** |
|  | 10 | 0.600 | 0.646 | 0.714 | **0.826** |
| CelebA-Hair | 0.5 | 0.497 | 0.592 | 0.513 | **0.663** |
|  | 1 | 0.498 | 0.606 | 0.474 | **0.675** |
|  | 5 | 0.469 | 0.609 | 0.508 | **0.680** |
|  | 10 | 0.462 | 0.626 | 0.540 | **0.671** |

Table 15: More classification results. Each value is derived by averaging the results from 50 independent trials.

|  | $\varepsilon$ | $n_{\text{critic}} = 1$ | $n_{\text{critic}} = 10$ | $n_{\text{critic}} = 50$ | $n_{\text{critic}} = 100$ | $n_{\text{critic}} = 200$ |
|---|---|---|---|---|---|---|
| Private-GANs | 0.5 | 0.468 | 0.448 | 0.612 | 0.620 | 0.598 |
|  | 1 | 0.570 | 0.541 | 0.629 | 0.663 | 0.498 |
|  | 5 | 0.584 | 0.572 | 0.652 | 0.680 | 0.634 |
|  | 10 | 0.607 | 0.673 | 0.677 | 0.714 | 0.623 |
| DPAF $(\mu = 1)$ | 0.5 | 0.490 | 0.650 | 0.511 | 0.463 | 0.515 |
|  | 1 | 0.507 | 0.666 | 0.491 | 0.466 | 0.549 |
|  | 5 | 0.535 | 0.690 | 0.500 | 0.524 | 0.496 |
|  | 10 | 0.503 | 0.666 | 0.552 | 0.452 | 0.546 |
| DPAF $(\mu = 8)$ | 0.5 | 0.501 | **0.668** | 0.579 | 0.500 | 0.517 |
|  | 1 | 0.567 | **0.725** | 0.573 | 0.537 | 0.510 |
|  | 5 | 0.687 | **0.715** | 0.524 | 0.511 | 0.503 |
|  | 10 | **0.789** | 0.690 | 0.556 | 0.519 | 0.472 |

Table 16: The classification accuracy of different $n_{\text{critic}}$'s on CelebA-Gender.

|  | $\varepsilon$ | $n_{\text{critic}} = 1$ | $n_{\text{critic}} = 10$ | $n_{\text{critic}} = 50$ | $n_{\text{critic}} = 100$ | $n_{\text{critic}} = 200$ |
|---|---|---|---|---|---|---|
| Private-GANs | 0.5 | 0.513 | 0.441 | 0.488 | 0.411 | 0.356 |
|  | 1 | 0.474 | 0.437 | 0.366 | 0.355 | 0.357 |
|  | 5 | 0.508 | 0.498 | 0.416 | 0.408 | 0.405 |
|  | 10 | 0.540 | 0.491 | 0.466 | 0.454 | 0.373 |
| DPAF $(\mu = 1)$ | 0.5 | 0.367 | 0.533 | 0.368 | 0.356 | 0.331 |
|  | 1 | 0.301 | 0.467 | 0.358 | 0.372 | 0.333 |
|  | 5 | 0.353 | 0.518 | 0.357 | 0.332 | 0.345 |
|  | 10 | 0.379 | 0.523 | 0.351 | 0.375 | 0.324 |
| DPAF $(\mu = 8)$ | 0.5 | **0.616** | 0.540 | 0.366 | 0.343 | 0.306 |
|  | 1 | 0.345 | **0.514** | 0.361 | 0.345 | 0.319 |
|  | 5 | 0.468 | **0.524** | 0.329 | 0.367 | 0.319 |
|  | 10 | **0.556** | 0.513 | 0.357 | 0.347 | 0.308 |

Table 17: The classification accuracy of different $n_{\text{critic}}$'s on CelebA-Hair.

### A.13 EMPIRICAL EVIDENCE FOR DATA PRIVACY OF DPAF AGAINST MIA

GAN-Leaks Chen et al. (2020b) offers a tool to evaluate whether a candidate GAN-synthesized dataset can resist MIA. Here, in addition to a formal privacy proof in Appendix A.6, we provide additional empirical evidence in Table 18 that the DPAF-synthesized dataset can resist MIA. In particular, each value in Table 18 refers to the AUC-ROC (area under the curve of ROC) of MIA in identifying members and non-members. One can see from Table 18 that if the synthetic dataset is generated by a model trained by ordinary SGD, then the AUC-ROC is 1, which means that MIA is always successful. On the other hand, in the case of the synthetic dataset from DPAF, the AUC-ROC is nearly 0.5, failing MIA.

| | | 64 Images | 128 Images |
|---|---|---|---|
| SGD | | 1.0 | 1.0 |
| DPAF | $\epsilon = 10$ | 0.466 | 0.502 |
| | $\epsilon = 5$ | 0.469 | 0.499 |
| | $\epsilon = 1$ | 0.473 | 0.499 |
| | $\epsilon = 0.5$ | 0.479 | 0.500 |

Table 18: AUC-ROC of MIA on CelebA-Gender with DPAF. The sizes of the training set is 64 and 128 (aligned with the setting used in Chen et al. (2020b)).