# OpenReview forum: "DPAF: Image Synthesis via Differentially Private Aggregation in Forward Phase"
_ICLR.cc/2024/Conference — ICLR 2024 Conference Withdrawn Submission_

### Official Review · Reviewer_yCfM · 2023-10-22

**Soundness:** 2 fair
**Presentation:** 1 poor
**Contribution:** 1 poor
**Rating:** 3
**Confidence:** 4

**Summary:**

This paper proposes a new framework for differentially private image synthesis, namely, DPAF, where the authors claim that they apply a DP feature aggregation in the forward phase for training a DP GAN. The training follows a two-stage paradigm: in the first stage, a classifier sharing the same architecture with the discriminator is pretrained with DP-SGD, where only a part of it is reused. In the second stage, only the discriminator (containing four parts: conv1,2,3 and FC) is trained with DP algorithm. Specifically, conv1 is transferred from the first stage; conv2 is trained with DP-SGD; there is a DPAGG between conv2 and 3 so that the subsequent conv3 and FC are also protected with DP guarantee.  Experiments show that it works in the high-dimensional image sets.

**Strengths:**

1. DP generative model is an active and important research field, and the generation in high-dimensional space is usually very challenging. This paper seems to target the high-dimensional RGB image sets, such as CelebA and FFHQ (up to 128x128).

**Weaknesses:**

1. The paper is poorly written, with too much unclarity in the main text. It is hard for me to follow and appreciate the design of DPAF. To name a few:
    + The caption of Figure 1 is not self-contained (yet it is supposed to be).
    + The Algorithm 1 gives details of the method but is included in appendix, which should be in the main text. In addition, section 3 and 4 should be rewritten, as they are too wordy in my opinion.
    + The most important question: what is the motivation of DPAGG? What is the comparison with DP-SGD? How and why does it reduce information loss (need either reference or experiment results)? Do you have the ablation study on DPAGG?
    + If you believe DPAGG can "reduce information loss", why not putting DPAGG between conv1 and conv2? Why conv2 needs to be trained with DP-SGD?
    + I don't get how eq.(2) is calculated. What is the explicit equation for the output of DPAGG? Why there is no adjacent input when you compute the sensitivity?
    + How large is $\sqrt{m}p$? How better it is compared to gradient sensitivity?
2. Since the discriminator has three parts with DP mechanisms, the authors need to allocate the total privacy cost $\epsilon$ for each part. However, the allocation strategy (in sec 4.2.4 and 4.2.5) looks heuristic and empirical. It is hard for me to believe the results presented are the optimal.
3. Results on grayscale image set is way worse than compared baselines. For example, in table 1, DPAF is more than 20% worse than the SOTA classification accuracy on MNIST and Fashion MNIST, which is a bit surprising, because MNISTs are easier datasets and are usually easier to get better results. Do authors have any interpretation on this result?
4. The title and abstract are misleading. Line 6 in Abstract says "Unlike previous methods, which add Gaussian noise in the backward phase during model training, DPAF adds differentially private feature aggregation in the forward phase...", which sounds like you do not have noise in the backward phase. However, DPAF has DP-SGD in the training too.

**Questions:**

I listed my questions above

---

### Official Review · Reviewer_NDSs · 2023-10-30

**Soundness:** 2 fair
**Presentation:** 2 fair
**Contribution:** 2 fair
**Rating:** 3
**Confidence:** 4

**Summary:**

This paper studies an important problem for differentially private image synthesis by proposing DP-AF. DP-AF introduces a module that adds noise in the forward pass instead of noise addition to gradients in DPSGD. Specifically, DP-AF splits privacy allocation into three parts: epsilon=epsilon_1+epsilon_2+epsilon_3. Firstly, DP-AF trains the earlier layers of the discriminator with the privacy budget  epsilon_1. DP-AF then jointly trains the discriminator and generator. Epsilon_2 is for train conv2 and epsilon_3 is for DPAGG.

**Strengths:**

- This paper studies an important problem, privacy-preserving image generation.
- The direction investigated in this paper, that is adding noise in the forward pass instead of noise adding in the gradients is interesting.

**Weaknesses:**

I have several concerns about the designs.

- It seems to me that Sec3.1 does not satisfy the epsilon_1 guarantee because conv2/conv3/FC is not trained in a privacy-preserving way. Although these layers will not be used in the follow-up steps, these layers contributed to the gradients that are backpropagated to conv1 and these layers parameters are updated based on private information. Therefore, it becomes intractable to bound the sensitivity for each private sample.
- The batch size choice is more constrained to the design. DP-trained generative models in general benefit from larger batch size [R1, R2]. However, the design of the aggregation layer limited the batch size. Although this work proposes the \mu to virtually increase the batch size, this is still a limit on batch size as shown in Table 8, while the larger batch size can help DP-SGD trained generative models. For example, [R2] uses a batch size of 16384.

In addition to the design concerns, I also think there is room for improvement in the writing
- Section 3 does not adequately motivate and validate the use of the aggregation layer and instead directly introduces the method. It would be better to briefly motivate the design in Section 3. Also, Section 3 could also be improved to be more clear and organized when introduce the design.
- The organization and takeaway of experiment results could be improved. For example, I currently feel kind of confused about the paragraph for “Random Parameters for conv1” and Section 4.2.5. “Random parameter for conv1” shows that an randomly initialized conv1 would not incur much degradation. Section 4.2.5 states intuitively that allocating privacy budgets is that the earlier (latter) layers should earn more (fewer) budgets. This indicates that more privacy budget allocated to conv1, but this is not consistent with the results in Table 7 and also not consistent with the results in “Random Parameters for conv1”.

Mino issues:

- Presentation issue:
  - Table number and caption should appear first instead of below.
  - Better to clearly state the default privacy allocation strategy, If I understand correctly, the canonical implementation is (0.1,  x, 0.1). It's better to clearly state this as canonical implementation instead of just explaining this example.
- Delta = 1e-5 is too high for the CelebA dataset, and it would be better to use delta=1e-6 for CelebA as in previous work[R1]. The comparison in this work is still fair though because this work conducts experiments on baselines for the same privacy parameters.


[R1] Alex Bie, Gautam Kamath, and Guojun Zhang. Private gans, revisited. Transactions on Machine Learning Research, 2023.

[R2] Sahra Ghalebikesabi, Leonard Berrada, Sven Gowal, Ira Ktena, Robert Stanforth, Jamie Hayes Soham De, Samuel L. Smith, Olivia Wiles, and Borja Balle. Differentially private diffusion models generate useful synthetic images. arXiv preprint arXiv:2302.13861

**Questions:**

- This work states that the architecture of G is a reverse of D.  I think it would be better if the authors could provide the network architecture details.
- How is the noise multiplier for epsilon_2 calculated ?
- Previous works also use FID to evaluate the quality of synthetic images. It would be better to conduct the FID study at least on one dataset.
- In Table 4, GelebaA-Gender at epsilon=10 in the (x, 0.2) setting is worse than epsilon=1 in the (x, 0.2). This is kind of counterintuitive as the former one has a relaxed privacy guarantee. I wonder if there is any brief explanation for this,
- It would be better to include the hyperparameter settings like in [R1, R2] like batch size, learning rate, steps, and clipping threshold.

---

### Official Review · Reviewer_xqsU · 2023-10-31

**Soundness:** 1 poor
**Presentation:** 2 fair
**Contribution:** 1 poor
**Rating:** 3
**Confidence:** 4

**Summary:**

The paper proposes Differentially Private Aggregation in Forward Phase (DPAF), a new approach for differentially private (DP) image generation. It first trains a feature extractor, and then trains a DP-GAN with private feature aggregation inserted in a middle layer. The paper demonstrates the privacy-utility trade-off of DPAF across multiple image datasets.

**Strengths:**

* The paper conducts comprehensive experiments and ablation studies across many datasets.

**Weaknesses:**

* The proposed approach violates DP. Please see the next section for the details.

**Questions:**

My major concern is that the proposed approach is NOT DP in both steps of the algorithm: classifier training and DP-GAN training.

Before talking about why the design in this paper is incorrect, let me first repeat the process of DP-SGD so that we can discuss it in the same language. Let $M_\theta$ be a neural network with parameter $\theta$, $x$ be a data point, and  $G(\theta, L(M_\theta(x)))$ be the gradient of the loss $L$ with respect to the parameter $\theta$ on $M_\theta(x)$. DP-SGD will update the parameter using $\frac{1}{N} (\sum_{i=1}^B CLIP(G(\theta, L(M_\theta(x_i))), C)+N(0,\sigma^2C^2I)$ where $\\{x_1,...,x_B\\}$ is a batch of samples, $CLIP(v, C)$ clips the gradient vector $v$ to norm C, and $\sigma$ is the noise multipler. This process is only DP under the following two conditions: (1) The loss $L(M_\theta(x_i))$ only depends on the sample $x_i$ and does NOT depend on the other samples $x_j, j\not=i$. Otherwise, even if we do clipping (i.e., $CLIP(\cdot, C)$), the sensitivity of each sample is not bounded. (2) The network weight $\theta$ involved in the computation of $G(\theta, L(M_\theta(x_i))$ (through previous iterations) is DP.

Now, let me explain why the proposed algorithm is NOT DP.

* Classifier training is not DP.

    (a) Firstly, the loss depends on all samples as stated in "C here is designed to predict the percentage of each class in a given batch". This violates condition (1).

    (b) Secondly, the fact that conv2 and conv3 are not trained through DP violates condition (2). More specifically, because conv2 and conv3 are not DP, the information from the past private samples seen by conv2 and conv3 can be leaked to conv1 through gradient computation and this data path is not guarded by any DP mechanism. From another perspective, even if the gradient clipping operation $CLIP(G(\theta, L(M_\theta(x_i))), C)$ makes sure that the contribution (sensitivity) of the sample $x_i$ in the **current** gradient computation for conv1 is bounded (in fact, it is not bounded due to point (a), but let's ignore it for now), $x_i$ can still contribute to future gradient computation of conv1 through conv2 and conv3.

* DP-GAN training is not DP.

    As conv3* and FC* are not trained with DP, and conv2* is trained with **non-DP** AGG, DP-GAN is not DP, which can be seen from the same reasoning as point (b) above.

I look at the proof in Appendix A.6. The key claim is that "As $\tilde{g}(\theta_1,D)$ is trained by DPSGD, one can ensure that $\tilde{g}(\theta_1,D)$ satisfy DP". This is only correct when conditions (1) and (2) are satisfied. Unfortunately, neither of them is satisfied with the proposed algorithm.

The paper also noted that methods such as GS-WGAN and G-PATE "have a similar design, where some parts of the model are trained by SGD but eventually discarded while the remaining parts are trained by DPSGD for the eventual release." While it is true that GS-WGAN and G-PATE also train part of the network with SGD, they use PATE-like aggregation mechanisms to ensure that the sensitivity from each private sample is strictly bounded and thus they satisfy DP. In the proposed algorithm, there is no such design, and as explained above, the sensitivity from each private sample is not bounded.

Because of this critical flaw, I have to give a rejection. Please clarify in the rebuttal if I misunderstood anything.